# Innovative Approaches for Maintaining and Enhancing Skin Health and Managing Skin Diseases through Microbiome-Targeted Strategies

**DOI:** 10.3390/antibiotics12121698

**Published:** 2023-12-04

**Authors:** Khadeejeh AL-Smadi, Vania Rodrigues Leite-Silva, Newton Andreo Filho, Patricia Santos Lopes, Yousuf Mohammed

**Affiliations:** 1Frazer Institute, Faculty of Medicine, The University of Queensland, Brisbane, QLD 4102, Australia; k.alsmadi@uqconnect.edu.au (K.A.-S.); vania.leite@unifesp.br (V.R.L.-S.); 2Departamento de Ciências Farmacêuticas, Instituto de Ciências Ambientais, Químicas e Farmacêuticas, Universidade Federal de São Paulo, UNIFESP-Diadema, Diadema CEP 09913-030, SP, Brazil; n.andreofilho@uq.edu.au (N.A.F.); patricia.lopes@unifesp.br (P.S.L.); 3School of Pharmacy, The University of Queensland, Brisbane, QLD 4102, Australia

**Keywords:** vitiligo, microbiome, prebiotics, probiotics, postbiotics

## Abstract

The skin microbiome is crucial in maintaining skin health, and its disruption is associated with various skin diseases. Prebiotics are non-digestible fibers and compounds found in certain foods that promote the activity and growth of beneficial bacteria in the gut or skin. On the other hand, live microorganisms, known as probiotics, benefit in sustaining healthy conditions when consumed in reasonable quantities. They differ from postbiotics, which are by-product compounds from bacteria that release the same effects as their parent bacteria. The human skin microbiome is vital when it comes to maintaining skin health and preventing a variety of dermatological conditions. This review explores novel strategies that use microbiome-targeted treatments to maintain and enhance overall skin health while managing various skin disorders. It is important to understand the dynamic relationship between these beneficial microorganisms and the diverse microbial communities present on the skin to create effective strategies for using probiotics on the skin. This understanding can help optimize formulations and treatment regimens for improved outcomes in skincare, particularly in developing solutions for various skin problems.

## 1. Introduction

The skin microbiome is a complex and dynamic ecology that resides in the skin, which is the largest organ of the human body. This varied group of microbes, which includes bacteria, fungi, and viruses, is essential to the homeostasis and health of the skin. A variety of skin diseases and disorders have been connected to disruptions of this critical microbial balance [1], such as psoriasis, vitiligo, and atopic dermatitis (AD) [2]. Promising new opportunities in the field of dermatology have recently arisen with the development of nanotechnology. Scientists and researchers have been investigating the development of innovative nanotechnology that provides efficacy in the restoration and control of the skin microbiome [3,4]. Nanoparticles can generally be utilized for delivering drugs in a way that improves their resistance to enzymatic degradation, targets specific locations, increases bioavailability, solubilizes them for intravascular transport, and maintains their effects [5]. These nanosystems or microbiome-targeted nanotherapeutics provide a novel approach that targets the related causes of skin disorders while reducing the adverse effects associated with traditional therapies.

The field of microbiome-targeted nanotherapeutics has been growing rapidly, and many nanosystems are being investigated for the regeneration and maintenance of the microbiota [6,7]. The nanostructures can be engineered to interact with the microbiome, either by modulating the microbial composition or delivering therapeutic agents that influence the microbiome, which affects the progression of certain skin conditions [8]. However, careful consideration of unexpected effects, biocompatibility, regulatory challenges, and long-term impacts is essential to ensuring the safety and efficacy of these innovative treatments [9]. Nanoparticles and nanoemulsions are the most popular nanosystems for targeting microbiomes because of their small size, modified properties, and efficient delivery capabilities. Their unique characteristics enable precise interactions with microbial communities and facilitate effective, customized interventions for improved dermatological treatments [3,10]. Nanoparticles typically range in size from 1 to 100 nanometers and are solid, colloidal particles. They can be made from a variety of materials, such as metals, polymers, and lipids. They include lipid nanoparticles (liposomes and solid lipid nanoparticles), polymeric micelles, metal nanoparticles, and polymeric nanoparticles (Figure 1). They are utilized in drug delivery since they may protect and deliver drugs to particular target locations within the body, enhancing medication effectiveness and minimizing adverse effects.

Furthermore, nanoparticles are employed to improve the performance of sunscreens, lotions, and other skincare products [11,12,13]. On the other hand, nanoemulsions are small oil droplets, typically ranging in size from 20 to 200 nanometers, distributed in an aqueous phase and frequently stabilized by surfactants in a process known as nanoemulsions formulations. They are either oil-in-water (O/W) or water-in-oil (W/O) nanoemulsions. The solubility and bioavailability of poorly water-soluble drugs can be enhanced by using nanoemulsions. In addition, they are used to improve the stability of some products as well as to encapsulate flavors, vitamins, and other bioactive components. In cosmetics, nanoemulsions are utilized in order to enhance the texture and stability of lotions and skincare items [14,15]. When creating medications in nanoform to increase absorption and effectiveness, three main variables need to be considered. These include FDA quality-related standards, drug transport, degradation mechanisms, and the stability of the manufactured drug. The pharmaceutical industry has limitations while utilizing nanoformulations because of this. The primary cause of the low therapeutic efficacy of nano-formulations is their propensity to self-aggregate at low drug concentrations, which compromises the formulation’s stability and increases the variability of drug entrapment because of polydispersity. Unformulated liposomes are more stable, but when a medicine is built into them, the stability decreases as the ionic strength grows. Starting with smaller nanoparticles (<20 nm) for the formulation could solve this issue such that even after loading the drug, it remains below 100 nm [16,17]. Although around 100 nm-sized nanoparticles have increased surface reactivity, they can have adverse biological effects like protein unfolding, membrane impairment, DNA damage, and inflammatory reactions. Additionally, several size-dependent processes, such as clathrin-mediated entry and caveolae-mediated pathways, allow particles in the 10–500 nm range, including those about 100 nm, to be absorbed into cells. Particle size preferences vary throughout cells; endothelial cells prefer particles that are around 100 nm in size, while professional phagocytes prefer larger particles [18,19].

Oxidative stress affects mitochondrial function and promotes the production of reactive oxygen species (ROS) and pro-inflammatory cytokines [20]. Increased ROS in the skin destroys melanocytes by damaging their DNA and associated cellular structures [21]. In addition, ROS causes a variety of oxidation products, including oxidized protein products and glycation products, that alter the structure of melanocytes [22]. It has been suggested that the main cause of melanocyte loss is mitochondrial malfunction driven by oxidative stress. Increased carbonylation in vitiligo melanocytes causes mitochondrial dysregulation, which could lead to severe mitochondrial dysfunction and melanocyte apoptosis [23]. The microbiome may influence the skin’s general durability and response to oxidative stress or contribute to the modulation of oxidative stress and inflammation [24].

In the exploration of skin health, emerging research underscores the role of the skin microbiome in various dermatological conditions. Among these conditions, vitiligo is an autoimmune condition characterized by depigmented macules and patches of various forms, which is caused by the death of melanocytes or the loss of their activity to produce natural skin pigments [25]. Consequently, discolored white spots appear on the skin, hair, and mucous membranes in various parts of the body. The prevalence of vitiligo in the general population of the world ranges from 0.06% to 2.28% [26]. Vitiligo can be classified into different types based on the characteristics of the lesions. Acrofacial vitiligo is characterized by lesions occasionally appearing on the extremities and face. Mixed vitiligo involves a combination of non-segmental and segmental patterns. Focal vitiligo is distinguished by small and isolated lesions. Mucosal vitiligo occurs in or around mucosal membranes. Universal vitiligo is a type where lesions develop and spread over the entire body [27,28]. Numerous mechanisms have been connected to the degeneration of melanocytes and the emergence of white patches in vitiligo. These mechanisms include neuronal, genetic, autoimmune, oxidative stress, environmental triggers, and the creation of inflammatory mediators [29]. The most prevalent and well-established explanation proposes that a defect in the immune response results in the destruction of melanocytes by autoimmune effector mechanisms, either memory cytotoxic T cells or autoantibodies directed against melanocyte surface antigens. Numerous studies have demonstrated a link between vitiligo and other autoimmune diseases such as Addison’s disease, pernicious anemia, diabetes, systemic lupus erythematosus, rheumatoid arthritis, psoriasis, and alopecia areata [30,31]. Topical corticosteroids (TCSs) and Topical calcineurin inhibitors (TCIs) are often used and recognized as a common treatment for many types of vitiligo [32]. When concerns regarding the reported adverse effects of prolonged usage of potent TCSs and TCIs arise, both have demonstrated a high degree of repigmentation but are not safe treatment options. TCIs often result in erythema, pruritus, and burning sensations as side effects, while TCSs increase the possibility of skin atrophy, striae, and telangiectasia [33,34]. Alternative treatments and outcomes have improved as a result of the introduction of numerous therapeutic approaches, such as phototherapy, excimer laser, vitamin D, and epidermal grafts [35,36]. However, there is occasionally hesitancy to suggest treatment because, historically, vitiligo was thought to respond to treatment quite poorly and infrequently [37]. Another common pathogenic trigger for melanocyte destruction in vitiligo patients is oxidative stress, which is also a key element in the onset and progression of the disease [38].

Psoriasis is a chronic inflammatory skin disease developed by the influence of a genetic predisposition and external variables. It has several different kinds; however, they are all identified by erythematous plaques that are frequently itchy [39]. The cutaneous immune system’s improper activation against a presumed infection is the main theory regarding the development of psoriasis. The innate and adaptive immune systems are both implicated in the pathogenesis of the disease, which also involves abnormal keratinocyte development and immune cell infiltration in the dermis and epidermis, with dendritic cells and T cells performing significant roles [40].

AD is the most prevalent inflammatory skin disorder worldwide. It affects a substantial portion of the global population, impacting around 15% to 20% of children and 1% to 3% of adults [41]. This chronic inflammatory skin condition is characterized by itchy and dry inflamed lesions. It stems from a range of hereditary mutations in skin barrier proteins and immunological defects specific to allergens. Individuals with AD often exhibit symptoms such as skin dryness, redness, and the development of erythematous lesions. Moreover, AD frequently presents with related issues like eczema, skin scaling, and persistent itching. Additionally, it can be associated with comorbidities such as allergic rhinoconjunctivitis and asthma [42].

Recently, evidence showed that altered skin microbiomes (dysbiosis) contribute to the pathogenesis of vitiligo by influencing immunological homeostasis, oxidative stress, and skin barrier [43,44]. Further, research on the skin lesions of psoriasis patients has revealed an imbalance in the skin microbiome. The authors report that inflammation results from a dysbiosis disruption of the skin’s immunological responses. *Streptococcus* bacteria are frequently overabundant, while *Cutibacterium* is less present in these lesions [39]. Studies are increasingly focusing on combining probiotics and nanotechnology to treat skin infections. Improved drug delivery, moisture retention, controlling the release of active compounds, and anti-infective properties have been demonstrated by nanotechnology, especially nanoparticle-based methods [45]. Some research suggests that topical probiotics emollient and bacteria-derived formulations may be an effective alternative for the treatment of AD [46]. Due to their capacity to have a favorable effect on the skin microbiome, these formulations can decrease inflammation and enhance the general health of the skin barrier by creating a balance in the microbial communities on the skin. [47]. Additionally, the controlled probiotic release offered by nanoparticles and nanotechnology delivers enormous potential for treating human infections. Although beneficial results have been observed, further human research is required to investigate their effectiveness in dermatological applications and to address the problem of bacterial resistance [48,49]. Therefore, in this review, we aim to provide a comprehensive evaluation of the use of topical probiotics, prebiotics, and postbiotics in treating skin disorders due to the increasing popularity of these topical treatments and the lack of clinical trials or efficacy studies to support their therapeutic value. Also, examine innovative approaches involving the use of microbiome-targeted techniques to maintain and improve overall skin health while addressing the management of different skin diseases.

## 2. Prebiotics, Probiotics and Postbiotics

Prebiotics are specific fermented substances or dietary supplements that are not digested, enhancing intestinal health by encouraging the growth of commensal bacteria [50]. On the other hand, probiotics are non-pathogenic live microorganisms, frequently yeast or bacteria, that, when administered in sufficient amounts, confer a health benefit to the host [51,52]. Probiotics from the first generation are commonly available products to treat microecological disorders. The next level of development is the production of “metabiotics”, which are small molecules or chemicals obtained from probiotic microorganisms. The bioactive compounds produced by symbiotic microorganisms (a combination of probiotics and prebiotics) from naturally occurring probiotic strains, or natural sources can be used to synthesize or semi-synthesize these metabiotics. They are known as “metabolic probiotics”, “postbiotics”, “biological drugs”, or “pharmacobiotics”. These compounds have the potential to impact the microbiota, human metabolic processes, signaling pathways, and physiological activities related to the host. Postbiotics have recognized chemical structures that may improve the composition and functionality of the host’s native microbiota as well as components involved in immunology, neurohormone biology, and metabolic and behavioral responses [53]. For instance, many commensal bacteria produce butyrate, a postbiotic that is a major source of energy for the colon and is essential for intestinal growth, differentiation, and inflammation control [54,55]. Moreover, postbiotics are probiotic-derived effector chemicals secreted by bacteria or released after lysis and capable of exerting qualities identical to those of the original probiotics [56,57] (Figure 2). They attempt to imitate the benefits of probiotics without taking the risk of administering live bacteria. While these probiotics, prebiotics, and postbiotics are commonly associated with dietary supplements targeting the gut microbiome, their application extends beyond the digestive system to various body parts, including the skin. In dermatology, particularly in addressing skin conditions, these concepts find relevance as researchers explore their potential in topical formulations and skincare products.

## 3. Skin Microbiome

The connection between microbial communities and the host tissue is symbiotic in the skin. The importance of resident microbial communities in maintaining the skin’s and immune system’s normal, healthy function has been demonstrated recently [58,59]. A diverse group of microorganisms known as the skin microbiome work together to maintain a complicated connection on the skin [1]. A large and diverse community of bacteria, viruses, and eukaryotes, including fungi and arthropods, comprise the human skin microbiota [60,61]. The heterogeneity of the skin microbiota, both in terms of its composition and prevalence, can be demonstrated in the significant variances between individuals and between different skin regions. These differences can be attributable to a complex interaction of elements, including genetic predisposition, dietary habits, choice of lifestyle, gender, age, ethnic background, and environmental circumstances [62,63]. The skin provides essential nutrients to establish its microbiota, including amino acids, fatty acids, and lactic acids from various sources like proteins, the stratum corneum, sweat, lipid hydrolysis, and sebum [64]. This relationship between the host and commensal microorganisms is vital for various physiological processes. To maintain this symbiotic relationship, commensal-specific T cells play a role in distinguishing between resident microorganisms and potential pathogens, thereby promoting tolerance towards the commensal microbiota [65]. The skin microbiome is made up of several different bacterial species. Microorganism imbalances can lead to skin diseases such as acne, AD, psoriasis, and rosacea [66]. Probiotic bacteria have been used to make a variety of treatments, nutritional items, and additives that support human health [67]. They provide several functions, one of which is serving as the first line of defense against invasive diseases and are also used to prevent both acute and chronic disorders or prophylaxis [68]. This shows that probiotics are used as an avoidance strategy for some disorders. In addition, specific health conditions, such as acute or chronic diarrhea and intestinal inflammation that can cause allergies, atherosclerosis, and cancer, are also treated with probiotic medications [69]. Coagulation-negative Staphylococci are also prevalent on human skin, and they act via a number of mechanisms including the epidermal barrier environment and the innate and adaptive immune systems found in the epidermis and dermis [70]. Further, the bacteriocins produced by this species have anti-inflammatory, and antibacterial characteristics that reduce the survival of harmful bacteria on the skin surface [41]. Endogenous urocanic acid found in the stratum corneum of the skin acts similarly to sunscreens in preventing damaging Ultraviolet (UV) radiation from penetrating the epidermis [67,71].

The microorganisms that make up the skin microbiome cooperate to keep the skin safe. However, due to many factors, such as external ones, commensal microorganisms may transform into pathogenic microbes, causing inflammation, itching, scaling, and other medical symptoms that point to an imbalance between our skin and its microbiome [72]. The word “dysbiosis” is used to describe how the microbiome of the skin has changed. Functional dysbiosis disturbs the interactions between bacteria and hosts and causes skin issues. Age, sex, hygiene, the use of particular pharmaceuticals, skin pH, sweating propensity, hair development on the skin, sebum production, usage of skin cosmetics, and lifestyle are only a few of the host factors that have an impact on the microbiome host interaction [73]. The potential of oral probiotics as a treatment for skin conditions has increased as research has revealed a connection between disrupted gut microbiota and inflammatory skin conditions [74]. Researchers are actively investigating the relationships among changes in the gut microbiota, immune system dysregulation, and the development or aggravation of autoimmune skin disorders, aiming to identify biomarkers and molecular pathways for potential therapeutic targeting [75,76,77]. Studies on probiotics have been performed using a concept known as the gut–brain–skin axis idea. These studies have demonstrated the efficacy of probiotics in the management of some dermatological conditions, including psoriasis, acne, vitiligo, and AD [2,78]. For example, AD is characterized by cutaneous dysbiosis and a greater presence of *Staphylococci* like *S. aureus* and *Malassezia* spp. These microorganisms release toxic chemicals and nanovesicles that trigger cytokines, which contribute to the persistence and aggravation of AD symptoms.

Transplanting specific strains of *S. epidermidis* and *S. hominis* that produce antimicrobial peptides resulted in significant decreases in the levels of *S. aureus* in individuals with AD. This implies a promising therapeutic strategy for addressing AD by influencing the skin’s microbiome to regulate and reduce the overgrowth of *S. aureus* [79]. The local skin microbiome plays a role in psoriasis pathogens. Psoriasis patients have a similar major species of bacteria in their skin flora compared to non-psoriasis individuals but with reduced diversity and changes in the relative abundance of certain bacteria. Specifically, lower concentrations of *Cutibacterium acnes* (formerly *Propionibacterium*) and *Actinobacteria* species are found in psoriasis patients, while higher concentrations of *Firmicutes*, *Proteobacteria*, *Acidobacteria*, *Schlegelella*, *Streptococcaceae*, *Rhodobacteraceae*, *Campylobacteraceae*, and *Moraxellaceae* species are observed when compared to controls. This suggests that alterations in the skin microbiome may contribute to psoriasis alongside immune system dysfunction [80,81]. As a result, these skin disorders present greater opportunities for probiotic research regarding topical benefits. In general, the gut microbiota is responsible for the body’s appropriate immunity and defense against harmful microbes. Therefore, alterations that are considered harmful at the intestinal microbiota level may result in infections and autoimmune diseases in a variety of organs outside of the colon, including the skin [82]. A recent study [83] shows that patients with vitiligo have a different microbial composition from healthy people, with a considerably lower Bacteroidetes to Firmicutes ratio. They also differ significantly from healthy people in 23 blood metabolites, and these metabolites are linked to particular microbial indicators. Commensal bacteria are vital components of the skin microbiome and play a crucial role in skin health. Another study [84] highlights that vitiligo-affected skin exhibits a dysbiosis in microbial community diversity, with lesional areas showing reduced taxonomic richness and evenness. Notably, Actinobacterial species are dominant in normal skin, while Firmicutes species dominate in vitiligo lesions, suggesting that these microbial changes could influence the development and severity of vitiligo.

## 4. Mechanisms of Action for Topical Prebiotics, Postbiotics and Probiotics

Many low-molecular-weight (LMW) bioactive substances, such as bacteriocins and other antimicrobial compounds, short-chain fatty acids, various fatty and organic acids, biosurfactants, polysaccharides, peptidoglycans, teichoic acids, lipo- and glycoproteins, vitamins, antioxidants, nucleic acids, amino acids, and different proteins, including enzymes and lectins, can be derived from different probiotic strains [85,86]. The applicable agents of these groups of LMW compounds isolated from symbiotic microorganisms or their cultural liquids may be used to produce functional foods, drugs for the prophylaxis and treatment of chronic human diseases, as well as sports and anti-aging foods [87,88]. The application of the probiotics concept in biotechnology has made it possible to include several thousand additional strains from the human-dominant intestinal phyla (Bacteroides, Firmicutes, Proteobacteria, Actinobacteria, and Archae) for nutritional and therapeutic purposes in addition to *Bifidobacteria*, *Lactobacilli*, *Escherichia*, and *Enterococci* sp. [89,90].

Microbiome development and changes are influenced by various factors such as childbirth, diet, drugs, and diseases [91]. The skin microbiota varies significantly across different body regions due to the presence of unique glands and hair follicles, creating distinct conditions for microbial growth. Specific bacterial and fungal species dominate various areas, such as lipophilic bacteria in sebaceous regions and fungal communities on the feet. Additionally, the facial skin microbiota is mainly composed of Proteobacteria, Firmicutes, Actinobacteria, and Bacteroidetes, with variations linked to age, and diversity differs by facial location, with cheek sites having the highest richness scores. Postbiotics like acetate, propionate, and butyrate play a crucial role in intestinal health by providing energy, enhancing the epithelial barrier, regulating immunity, and preventing pathogen invasion [92]. Immune diseases, inflammation, and gut dysbiosis can result from a dysregulation of this balance [93]. The gut–skin axis is proposed as a connection between emotional states, gut health, and skin conditions. Increased intestinal permeability can activate T cells, disrupt immunosuppressive factors, and lead to systemic inflammation, potentially affecting skin homeostasis [94]. Gut microbes can also communicate with other organs through neurotransmitter production. Therefore, changes in the gut microbiome may directly impact systemic inflammation [92]. In the last century, the ability to identify microorganisms based on their appearance or biochemical traits and improvements in cell culture techniques have allowed for the expansion of study into the microbial variety of human skin. Researchers have identified numerous genera of bacteria that are typically found on healthy skin using culture-dependent methods. These genera include *Staphylococci*, *Micrococci*, *Corynebacteria*, *Brevibacteria*, *Propionibacteria*, and *Acinetobacter* [95]. *Staphylococcus aureus*, *Streptococcus pyogenes*, and *Pseudomonas aeruginosa* were identified at the species level using culture techniques, such as colonizers in unusual conditions [96]. The skin microbiome is primarily made up of two main types of bacteria: the resident and transient microbiota types. The resident microbiota is the most significant and persistent group and may regenerate after any disturbances [97]. In contrast, the transitory microbiome is environment-dependent and only stays on the skin for a few hours or days [98]. Both of these microbiota types are harmless in healthy skin. Actinobacteria, Firmicutes, Proteobacteria, and Bacteroides are some of the most prevalent phyla on the skin, while the most common genera are *Corynebacterium*, *Propionibacterium*, and *Staphylococci* [66,99].

## 5. Clinical Verification and Effectiveness

The effectiveness of probiotic products used topically has received very little investigation. However, in the past ten years, the number of commercially available topical probiotics has dramatically increased [100], and probiotics have been applied topically and orally to treat various skin disorders [53] Table 1. The gut microbiota significantly affects the immune system, and many studies [2,101,102] have shown the importance of dysregulations in the skin and gut microbiome in immune-related diseases. Immune dysregulation driven by imbalances in the gut microbiota may involve an excessive immune response that targets melanocytes and contributes to their destruction in vitiligo [103]. It is thought that both genetic and environmental factors play a role in the vitiligo development process. Recent research indicates that vitiligo patients have altered immune responses and increased stress-induced production of Interferon-gamma (IFN-γ), which leads to melanocyte apoptosis [104]. Dysbiosis in the gut microbiome is observed in vitiligo patients, with reduced Bacteroides populations and changes in microbiota diversity, which are associated with mitochondrial damage and peripheral changes in innate immunity [105]. Skin microbiota composition also differs between vitiligo patients and healthy controls, particularly in vitiligo lesions, where there is a reduction in *Staphylococcus* and *Cutibacterium* and an increase in *Proteobacteria* associated with inflammation [103].

The topical application of probiotic bacteria may help enhance the skin’s natural barrier by directly affecting the site of application. This may be performed by the resident bacteria and the probiotic bacteria that produce certain antimicrobial amino peptides that benefit the immune responses in the skin and help eliminate pathogens [106]. The administration of probiotic species that are not native to a particular ecosystem can potentially cause adverse effects [107]. They are commonly included in over-the-counter cosmeceutical products, but their effectiveness may be compromised by their high bacterial load and the preservatives used, which can impact the skin’s microbiota [108]. Probiotics have been utilized in a variety of cosmetic items, including lotions, intimate hygiene products, shampoos, and toothpaste. These strains include *Bacillus subtilis*, *Lactobacillus acidophilus*, *Lactobacillus casei*, and *Lactobacillus plantarum*. These probiotics provide several benefits for skin health, including moisturizing effects, reducing toxic metabolites, enhancing antibody production, restoring immune system balance, and regulating cytokine synthesis [109,110]. In addition, topically applied probiotics can serve as a protective barrier on the skin by competing with and inhibiting the binding of potential pathogens to skin sites. This competitive inhibition helps prevent the colonization of harmful microorganisms on the skin, further contributing to skin health and protection [100].

Postbiotics, formed from microbial growth by-products or inactive dead strains, positively benefit skin health because they contain bioactive substances such as bacteriocins, lipoteichoic acids, and organic acids [111]. Species of *Lactobacillus*, including those found in cosmetics, create lactic acid, which aids in moisturization and anti-aging [112]. *Streptococcus* and *Bifidobacterium* strains have also been demonstrated to increase skin hydration and elasticity, with *Bifidobacterium* contributing to the production of hyaluronic acid for improved skin appearance [113,114].

Novel strategies in the field of dermatology employ nanocarriers to improve the topical applications of probiotics and prebiotics [4,6]. These nanocarriers, often in the form of nanoparticles or nanoemulsions, serve as efficient vehicles for loading and delivering probiotic strains to the skin. This advanced delivery system not only protects the viability of probiotics during formulation but also enhances their penetration into the skin, maximizing their potential to establish a protective barrier.

## 6. Formulations and Delivery Methods

Skin microbiota can be preserved and restored by probiotics, prebiotics, or combination supplements (symbiotic) [106] Table 2. In other words, probiotics, prebiotics, and symbiotics are the three treatment modalities currently employed to maintain and restore the gut microbial ecology [90,115]. In cosmetic formulations, prebiotics can selectively increase the activity and growth of beneficial skin probiotics [116]. Some cosmetic formulations may help foster the normal skin microbiome by being selective in their activity [106]. It has recently been demonstrated that incorporating antioxidants and probiotics in combination with other therapies can accelerate repigmentation. One study on AD skin showed that topical formulations with particular probiotic strains helped reduce skin lesions [117]. Another study conducted by Chaudhry et al. [118] combined TCI with an antioxidant and probiotic diet as an adjuvant. They demonstrated that full repigmentation could be achieved in just six weeks and that no relapses were noted for 52 weeks following the treatment. Certain lactic acid bacteria, specifically *Streptococcus thermophiles*, have been found to enhance ceramide production in the skin when topically applied as a cream. Ceramides are lipids that play a crucial role in maintaining the skin’s barrier function and hydration levels, and this is particularly beneficial for individuals with acne-prone skin because acne treatments can sometimes lead to dryness and irritation. Additionally, it shows antimicrobial activity against *Cutibacterium acnes*, a bacterium associated with the development of acne [119].

Incorporating live bacteria, such as probiotics, into cosmetic products is a complex process that requires significant adjustments in production, storage, and distribution procedures [120]. These adjustments are necessary because formulation processes can potentially deactivate probiotics and alter their intended functions. Different dehydration techniques are frequently employed in probiotic manufacture to overcome this [121]. In contrast to oral probiotic formulations, topical probiotic products often need reconstitution in a vehicle, such as creams, gels, and emulsions, before use, allowing them to be applied to the skin and integrated into specific pharmaceutical bases [122,123] Figure 3. Additionally, many factors can influence the probiotic quality during storage or delivery because of their susceptibility to temperature, humidity, and cooling conditions [124]. Prebiotics and postbiotics have recently been proposed as alternative options to address these concerns.

Prebiotics or symbiotics have been demonstrated to improve skin health significantly. For instance, Lactocare^®^, a symbiotic composition, combined with topical hydrocortisone improved psoriasis scores when administered orally [125]. A different study showed that giving mice with AD olive-derived antioxidant dietary fiber caused enhancements in the gut microbiota’s composition, cytokine profiles, and butyrate synthesis, which improved the immune system response linked to AD [126]. Prebiotic polysaccharides such as fructo-oligosaccharides, galacto-oligosaccharides, and milk-derived oligosaccharides are fermented by bacteria (probiotics) in the human colon, resulting in the production of short-chain fatty acids (SCFAs), such as acetate, propionate, and butyrate [127,128,129]. These SCFAs increase blood flow in the colon, improve fluid and electrolyte absorption in the gut, promote the growth of intestinal cells, decrease bowel inflammation, influence enzyme activity, and reduce the risk of cancer and pathogen colonization [130,131]. Intestinal cells use butyrate as an energy source, which stimulates proliferation, increases the production of protective mucin, strengthens the intestinal barrier, and enhances immune system performance [129,131]. The human gut contains a variety of strains from the *Lactobacillus* and *Bifidobacterium* genera, which have a range of advantageous benefits. Additionally, these bacteria are used to generate probiotics [127].

Postbiotics produced from several probiotic strains have demonstrated potential therapeutic advantages for various skin issues and disorders. Cell-free supernatants, lysates, and bioactive peptides are some examples of postbiotics that have been examined for their effects on skin health and have shown a variety of beneficial results [132]. Postbiotics from *Lactobacillus fermentum*, *Lactobacillus reuteri*, and *Lactobacillus subtilis* natto are being investigated as a novel method to promote earlier full epithelization and decrease skin inflammation when applied topically with a cold cream [133]. A cell-free supernatant of fermented milk from the *L. helveticus* strain demonstrated substantial antioxidant capabilities against UVB-induced skin damage, including decreasing lipid peroxidation and inhibiting melanin formation [132]. A customized blend of probiotic strains (*Lactobacillus plantarum*, *Lactobacillus casei*, and *S. thermophilus*) has shown promise in reducing pore size and wrinkle depth, as well as maintaining hydration and skin smoothness [134]. *S. thermophiles* lysate with sphingomyelinase has been demonstrated to strengthen the lipid barrier of the skin, elevate stratum corneum ceramide levels, and lower water loss [135]. In order to stimulate the expression of moisturizing factors in the skin, *L. plantarum* ferment lysate and *L. plantarum* K8 strain lysate have been proposed as functional ingredients for moisturizing cosmetics [136]. Improvements in acne lesions have been seen, along with decreases in transepidermal water loss and sebum production, when using *L. plantarum* ferment lysate [137]. Multiple positive effects of *Vitreoscilla filiformis* lysate on skin health have been observed, including modulation of skin immunity, inflammation reduction, and skin barrier enhancement [138]. LactoSporin^®^ formulation, containing cell-free supernatants of *Bacillus coagulans* and inactivated cells of *Bacillus longum*, has been effective in managing mild-to-moderate acne lesions and seborrheic conditions, surpassing benzoyl peroxide in some cases [139]. A skincare cream containing fermented lysate derived from Lactobacillus plantarum can effectively and safely cure mild-to-moderate acne vulgaris after four weeks of topical treatment [140]. By potentially enhancing the skin barrier and regulating the skin’s immune system, a postbiotic derived from *Bifidobacterium lactis* has shown effectiveness in reducing dandruff [141]. A topical formulation containing plantaricin A bioactive peptides, postbiotics, and *Lactobacillus kunkeei* isolated from bee bread, along with *Tropaeolum majus* flower/leaf/stem extract, has shown significant improvement in patients with alopecia areata [142] Figure 4.

## 7. Challenges in Maintaining Probiotic and Prebiotic Viability and Stability

Microbiomes can directly protect the host against pathogens, manage inflammation, and modify the functions of the adaptive immune system [143]. Microbial populations within the human body differ across various body sites. These variations can be influenced by the immune system, environment, and interactions between various microbial species [144]. The potential health benefits of probiotics have sparked interest in developing dermo-cosmetics derived from them. However, there are significant technological and legal challenges in using live microbes on the skin [145].

Furthermore, given that many people use cosmetics daily, these products can influence the skin microbiota. While cosmetics do not need to be completely sterile, they must adhere to current regulatory standards to ensure they do not carry harmful bacteria. Nevertheless, the inclusion of antimicrobial preservatives in formulations may affect the skin microbiome, potentially leading to long-term consequences such as the emergence of antibiotic-resistant bacteria [146]. In recent research, biomolecules such as polysaccharides (chitosan, xanthan, dextrin, and carrageenan) and milk proteins have been used to microencapsulate probiotics and other bioactive components. Sodium alginate is used more because of its cheapness and easy availability [147]. Therefore, it is important to keep in mind that it can be challenging to create and maintain the stability of probiotics in skincare products. Probiotics’ viability and efficacy can be affected by variables such as pH, temperature, and exposure to air and light. Thus, while developing a formulation for topical treatment, it is crucial to ensure that the selected probiotic strains can maintain their viability and activity within the formulation.

## 8. Conclusions

The significance of microbiome restoration as a potential therapeutic strategy is highlighted by the skin microbiome’s role in preserving skin homeostasis and its connections to numerous skin disorders. The possible use of probiotics, prebiotics, and postbiotics in topical preparations for many skin disorders is now the subject of investigation and study. Prebiotics contribute to improved gastrointestinal well-being by exhibiting anti-inflammatory and immunomodulating effects. They prevent the colonization of harmful microorganisms, support the integrity of the intestinal barrier, and enhance overall immune function, which can be considered a novel alternative therapy for many immune-related skin diseases. Postbiotics derived from probiotic strains and bacterial lysates have shown promise in various aspects of skin health, including moisturization and inflammation reduction. New techniques may employ nanocarriers to improve probiotic and prebiotic topical administration by protecting their viability, enabling effective loading, and enhancing skin penetration. These findings suggest the potential for postbiotic-based skincare products and treatments in addressing a wide range of dermatological concerns. However, it is essential to acknowledge the challenges ahead. Regulatory considerations and safety concerns surrounding the use of microbiome in skincare require careful attention. While oral probiotics have a clear legal definition tied to bacteria and yeast, the lack of such a definition for skin probiotics underscores a regulatory gap, signaling a pressing need for guidelines in this rapidly advancing field. Further research is also needed to understand their long-term effects and optimal formulations.

## Figures and Tables

**Figure 1 antibiotics-12-01698-f001:**
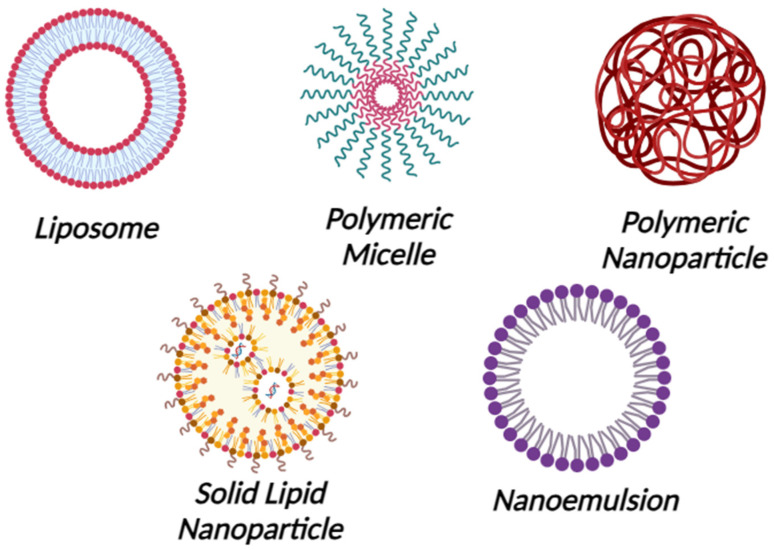
Nanoparticles and Nanoemulsions. Created with Biorender.com.

**Figure 2 antibiotics-12-01698-f002:**
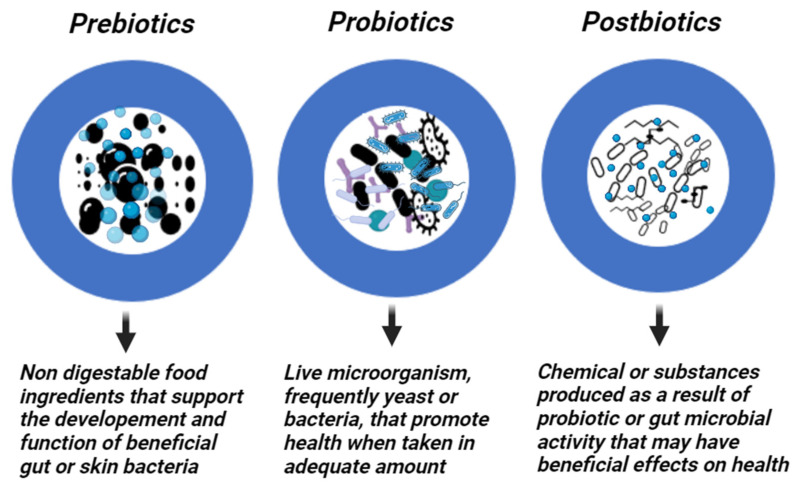
Prebiotics, Probiotics, and Postbiotics. Created with Biorender.com.

**Figure 3 antibiotics-12-01698-f003:**
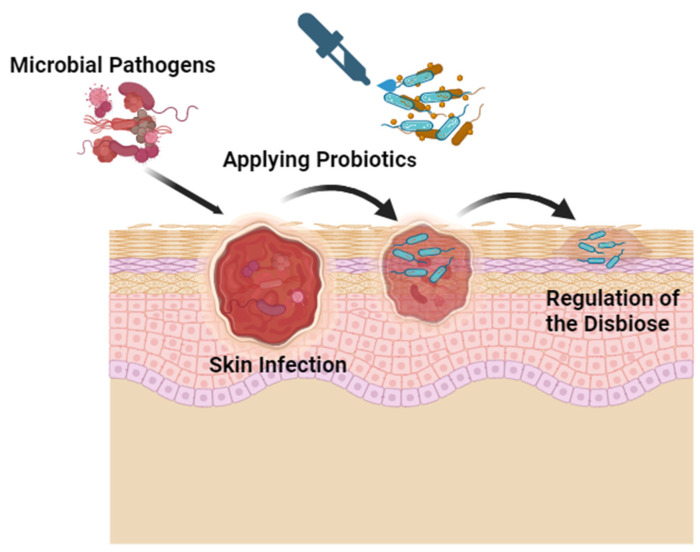
Probiotics Application During Skin Infection. Created with Biorender.com.

**Figure 4 antibiotics-12-01698-f004:**
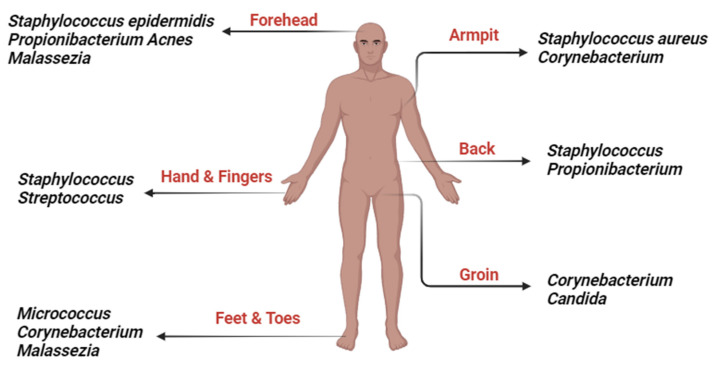
Skin Microbiome Diversity at Different Body Regions that can be Targeted. Created with Biorender.com.

**Table 1 antibiotics-12-01698-t001:** Examples of Probiotic-Containing Commercial Products.

Probiotics	Oral	Topical	Benefits Claimed
*Lactobacillus*			
		*Lactobacillus Ferment* Essence	A skincare brand containing *Lactobacillus ferment* for skin nourishment
	Probiotic Complex with *Lactobacillus Acidophilus*		a health brand promoting overall well-being, including potential benefits for the skin
*Bifidobacterium*			
		*BifidoBalance* Cream	A skincare company formulated with *Bifidobacterium* to support skin microbiome balance
	Gut Health Probiotic Blend with *Bifidobacterium*		A nutritional supplement brand aimed at promoting gut health with potential skin benefits
*Streptococcus thermophilus*			
		Thermal Probiotic Cream	Skincare line featuring *Streptococcus thermophilus* for enhancing the diversity of the skin microbiome
*Saccharomyces boulardii*			
	*Saccharomyces boulardii* Probiotic Capsules		A wellness brand specifically designed to support gut health and potentially improve skin conditions
Probiotic Blends			
		Probiotic Power Serum	A skincare brand incorporating a blend of *Lactobacillus*, *Bifidobacterium*, and *Streptococcus thermophilus* for a comprehensive skin health approach
	Daily Probiotic Blend Capsules		A health and wellness company offering a mix of various probiotic strains for overall health, including potential benefits for the skin

**Table 2 antibiotics-12-01698-t002:** Overview of Formulations and Delivery Methods in Skin Microbiota Preservation.

Treatment Modality	Delivery Method	Key Components	Results and Applications
Probiotics	Topical application	Live bacteria (Probiotics)	Reduction of skin lesions in AD; potential for repigmentation
Prebiotics	Oral administration	Substances promoting beneficial bacteria growth (e.g., fructooligosaccharides)	Improved psoriasis scores when combined with topical hydrocortisone; positive effects on gut microbiota linked to AD
Symbiotics	Oral administration	Combination of probiotics and prebiotics (e.g., Lactocare^®^)	Improved psoriasis scores when used alongside topical hydrocortisone
Postbiotics	Topical application	Cell-free supernatants, lysates, bioactive peptides	Acceleration of epithelization, reduction of skin inflammation, antioxidant capabilities against UVB-induced damage; multiple positive effects on skin health

## Data Availability

Not applicable.

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
