# Peer review of "Innovative Approaches for Maintaining and Enhancing Skin Health and Managing Skin Diseases through Microbiome-Targeted Strategies"

_antibiotics, 2023, doi:10.3390/antibiotics12121698_

Round 1
Reviewer 1 Report
Comments and Suggestions for Authors
This review article tries to discuss the various approaches to maintaining skin functioning by management of skin health through the targeted strategies.
This review article perfectly presents the content and the application of probiotics, prebiotics, and postbiotics with proper mechanisms of action.
1. From Lines no.41 to 68, the authors discussed the advantages and uses of nanoformualtion with the particle size. However, the authors should also mention the limitation of nanoformulation in terms of particle size. If the particle size of any topical formulations is around 100 nm, they could enter the blood circulation. So, authors should also write about possible limitations of nanoformulations topical drug delivery approach.
2. From Lines no. 315 to 317, the authors discussed that “Although in the past ten years, the number of commercially available topical probiotics has dramatically increased [96], and probiotics have been applied topically and orally to treat a variety of skin disorders.”
So, in support of this statement, authors should also add the table of a probiotic-containing commercial product.
Author Response
Response to Reviewers
We would like to thank the editor and reviewers for their time and effort in reading and commenting on our paper. The constructive comments have been considered, and the manuscript has been updated accordingly. The responses to the reviewers' comments are listed below, and they are highlighted throughout the manuscript.
Reviewer 1
- Comment 1- From Lines no.41 to 68, the authors discussed the advantages and uses of nanoformulation with the particle size. However, the authors should also mention the limitation of nanoformualtion in terms of particle size. If the particle size of any topical formulations is around 100 nm, they could enter the blood circulation. So, authors should also write about possible limitations of the nanoformualtion topical drug delivery approach.
- Response: We appreciate your feedback and would like to draw your attention to (section Introduction) Lines 68-86 of the article where we extensively covered the limitations of nanoformulations regarding size and stability. Below details have been added to the introduction of the manuscript and the reference list, also it was highlighted in the manuscript:
When creating medications in nanoform to increase absorption and effectiveness, some variables need to be considered. These include FDA quality-related standards, drug transport and degradation mechanisms, and the stability of the manufactured drug. The pharmaceutical industry has limitations while utilising nanoformulations because of this. The primary cause of the low therapeutic efficacy of nano-formulations is their propensity to self-aggregate at low drug concentrations, which compromises the formulation's stability and increases the variability of drug entrapment because of polydispersity. Unformulated liposomes are more stable, but when a medicine is built into them, the stability decreases as the size of the ionic strength grows. Starting with smaller nanoparticles (<20 nm) for the formulation could solve this issue such that even after loading the drug it remains below 100nm [16, 17]. Although around 100 nm-sized nanoparticles have increased surface reactivity, they can have adverse biological effects like protein unfolding, membrane impairment, DNA damage, and inflammatory reactions. Additionally, there are several size-dependent processes, such as clathrin-mediated entry and caveolae-mediated pathways, that allow particles in the 10-500 nm range, including those that are about 100 nm, to be absorbed into cells. Particle size preferences vary throughout cells; endothelial cells prefer particles that are around 100 nm in size, while professional phagocytes prefer larger particles [18, 19].
References:
- 16. Choi, S.H., et al., Thermally reversible pluronic/heparin nanocapsules exhibiting 1000-fold volume transition. Langmuir, 2006. 22(4): p. 1758-1762.
- 17. Barenholz, Y.C., Doxil®—The first FDA-approved nano-drug: Lessons learned. Journal of controlled release, 2012. 160(2): p. 117-134.
- 18. Akinc, A. and G. Battaglia, Exploiting endocytosis for nanomedicines. Cold Spring Harbor perspectives in biology, 2013. 5(11): p. a016980.
- 19. Nel, A., et al., Toxic potential of materials at the nanolevel. science, 2006. 311(5761): p. 622-627.
- Comment 2- From Lines no. 315 to 317, the authors discussed that “Although in the past ten years, the number of commercially available topical probiotics has dramatically increased [96], and probiotics have been applied topically and orally to treat a variety of skin disorders.”
So, in support of this statement, authors should also add the table of a probiotic-containing commercial product.
- Response: We would like to thank the reviewer for the valuable comment, we have incorporated the suggested information about adding a table of a probiotic-containing commercial product. Below details have been added to the section on Clinical Verification and effectiveness of the manuscript.
- Lines 378-379: Table 1 examples of a probiotic-containing commercial product
-
· Probiotics
· Oral
· Topical
· Benefits Claimed
· Lactobacillus
·
·
·
·
·
· Lactobacillus Ferment Essence
· A skincare brand, containing Lactobacillus ferment for skin nourishment
·
· Probiotic Complex with Lactobacillus Acidophilus
·
· a health brand, promoting overall well-being, including potential benefits for the skin
· Bifidobacterium
·
·
·
·
·
· BifidoBalance Cream
· A skincare company, formulated with Bifidobacterium to support skin microbiome balance
·
· Gut Health Probiotic Blend with Bifidobacterium
·
· A nutritional supplement brand, aimed at promoting gut health with potential skin benefits
· Streptococcus thermophilus
·
·
·
·
·
· Thermal Probiotic Cream
· Skincare line, featuring Streptococcus thermophilus for enhancing the diversity of the skin microbiome
· Saccharomyces boulardii
·
·
·
·
· Saccharomyces Boulardii Probiotic Capsules
·
· A wellness brand, specifically designed to support gut health and potentially improve skin conditions
· Probiotic Blends
·
·
·
·
·
· Probiotic Power Serum
· A skincare brand, incorporating a blend of Lactobacillus, Bifidobacterium, and Streptococcus thermophilus for a comprehensive skin health approach
·
· Daily Probiotic Blend Capsules
·
· A health and wellness company, offering a mix of various probiotic strains for overall health, including potential benefits for the skin
Reviewer 2 Report
Comments and Suggestions for Authors
This review reports that the skin microbiome plays a crucial role in maintaining skin health, and disruptions in this microbial balance are associated with various skin diseases. Prebiotics, found in certain foods, promote the growth of beneficial bacteria in the skin, while probiotics contribute to sustaining healthy conditions. Distinguishing between live microorganisms (probiotics) and their by-product compounds (postbiotics), the review highlights the significance of understanding the dynamic relationship between these beneficial microorganisms and diverse microbial communities on the skin. By exploring novel strategies for microbiome-targeted treatments, this review underscores the potential to optimize formulations and treatment regimens, offering effective solutions for maintaining and enhancing overall skin health while managing various dermatological conditions. The only correction needed is changing “fibre” to “fiber” in Line 13 and 396.
Author Response
Reviewer 2
This review reports that the skin microbiome plays a crucial role in maintaining skin health, and disruptions in this microbial balance are associated with various skin diseases. Prebiotics, found in certain foods, promote the growth of beneficial bacteria in the skin, while probiotics contribute to sustaining healthy conditions. Distinguishing between live microorganisms (probiotics) and their by-product compounds (postbiotics), the review highlights the significance of understanding the dynamic relationship between these beneficial microorganisms and diverse microbial communities on the skin. By exploring novel strategies for microbiome-targeted treatments, this review underscores the potential to optimize formulations and treatment regimens, offering effective solutions for maintaining and enhancing overall skin health while managing various dermatological conditions.
Comment 1 The only correction needed is changing “fibre” to “fiber” in Lines 13 and 396.
- Response: We value the reviewer's accurate suggestion and the words have been modified. (Line 13 and 423).
Reviewer 3 Report
Comments and Suggestions for Authors
The authors have nicely compiled the data. However following comments need to be addressed for enhancing the quality of the manuscript.
1. An error was observed related to the in-text citation of the reference. Check the whole manuscript once, for its betterment.
2. Manuscript quality can be enhanced by incorporating the table, which represents the data of high importance in relation to the title of the manuscript, in the section "Formulations and Delivery Methods"
3. The manuscript needs to be checked once again with respect to the grammatical errors.
Comments on the Quality of English LanguageThe manuscript needs to be checked once again with respect to the grammatical errors.
Author Response
Reviewer 3
The authors have nicely compiled the data. However following comments need to be addressed for enhancing the quality of the manuscript.
Comment 1- An error was observed related to the in-text citation of the reference. Check the whole manuscript once, for its betterment.
- Response: Thank you for your critical feedback, we have corrected the cross references for the two figures: Figure 1 Line 57 and Figure 2 Line 196.
Comment 2: Manuscript quality can be enhanced by incorporating the table, which represents the data of high importance in relation to the title of the manuscript, in the section "Formulations and Delivery Methods".
- We value the reviewer’s perceptive comment. We have incorporated the suggested information regarding incorporating the table, which represents the data of high importance in relation to the title of the manuscript, in the section "Formulations and Delivery Methods.
- Line 402-403 TABLE 2 Overview of Formulations and Delivery Methods in Skin Microbiome Preservation
Treatment Modality |
Delivery Method |
Key Components |
Results and Applications |
Probiotics |
Topical application |
Live bacteria (Probiotics) |
Reduction of skin lesions in atopic dermatitis; potential for repigmentation |
Prebiotics |
Oral administration |
Substances promoting beneficial bacteria growth (e.g., fructo-oligosaccharides) |
Improved psoriasis scores when combined with topical hydrocortisone; positive effects on gut microbiota linked to atopic dermatitis |
Symbiotics |
Oral administration |
Combination of probiotics and prebiotics (e.g., Lactocare®) |
Improved psoriasis scores when used alongside topical hydrocortisone |
Postbiotics |
Topical application |
Cell-free supernatants, lysates, bioactive peptides |
Acceleration of epithelization, reduction of skin inflammation; antioxidant capabilities against UVB-induced damage; multiple positive effects on skin health |
Comment 3 The manuscript needs to be checked once again with respect to the grammatical errors.
- Response: We appreciate the reviewer's insightful feedback, regarding the grammatical errors all have been corrected according to reviewer suggestions.
Reviewer 4 Report
Comments and Suggestions for Authors
Authors conducted a review which explores novel strategies that use microbiome-targeted treatments to maintain and enhance overall skin health while managing various skin disorders. The article explores a relevant and rapidly advancing field, contributing to the understanding of the skin microbiome and its potential applications in skincare. Introduction provides a comprehensive overview of the skin microbiome and its connection to various skin diseases. However, some sentences are very complex and could be simplified for be"er clarity.English level has good quality. In lines 56-57, 176-177 should be a reference.Provide full words then abbreviations. Develop visuals to illustrate how probiotics work on the skin.Please create an infographic that visually represents the diverse microbial communities on the skin, highlight different skin regions and their unique microbiota compositions.
Author Response
Reviewer 4
Authors conducted a review which explores novel strategies that use microbiome-targeted treatments to maintain and enhance overall skin health while managing various skin disorders. The article explores a relevant and rapidly advancing field, contributing to the understanding of the skin microbiome and its potential applications in skincare. An introduction provides a comprehensive overview of the skin microbiome and its connection to various skin diseases. However, some sentences are very complex and could be simplified for be"er clarity. English level has good quality.
Comment 1) In lines 56-57, 176-177 should be a reference.
- Response: We would like to thank the reviewer for the valuable comments, and we have corrected the cross-references for the two figures: Figure 1 Line 57 and Figure 2 Line 196.
- Comment 2) Provide full words and then abbreviations.
- Response: Thank you for your critical feedback and all abbreviations have been provided with full words (Lines 142, 236, and 341).
- Comment 3) Develop visuals to illustrate how probiotics work on the skin.
- Response: We appreciate the reviewer's precise recommendation, and we have created a visual to illustrate how probiotics work on the skin: Figure 3 Probiotics Application During Skin Infection
Line 415-416.
- Comment 4) Please create an infographic that visually represents the diverse microbial communities on the skin, highlight different skin regions and their unique microbiota compositions.
- Response: We appreciate the reviewer's insightful feedback, and we have created an infographic that visually represents the diverse microbial communities on the skin, highlights different skin regions and their unique microbiota compositions: Figure 4 Figure 4 Skin Microbiome Diversity at Different Body Regions that can be Targeted
Line 467-470.